# Machine learning evaluation for identification of M-proteins in human serum

**Alexandros Sopasakis[1], Maria Nilsson[2], Mattias Askenmo[2], Fredrik Nyholm[2], Lillemor Mattsson Hultén[2,3], Victoria Rotter Sopasakis** [2,4] *

1 Mathematics, Faculty of Engineering (LTH), Lund University, Lund, Sweden, 2 Department of Clinical Chemistry, Region Västra Götaland, Sahlgrenska University Hospital, Gothenburg, Sweden, 3 Department of Molecular and Clinical Medicine, Wallenberg Laboratory, Institute of Medicine, Sahlgrenska Academy, University of Gothenburg, Gothenburg, Sweden, 4 Department of Laboratory Medicine, Institute of Biomedicine, Sahlgrenska Academy, University of Gothenburg, Gothenburg, Sweden

* victoria.rotter@clinchem.gu.se

**Data Availability Statement:** No - some restrictions will apply; The data have been uploaded in a public data repository, the Swedish National Data Service (https://snd.gu.se/en), with

## Abstract

Serum electrophoresis (SPEP) is a method used to analyze the distribution of the most important proteins in the blood. The major clinical question is the presence of monoclonal fraction(s) of antibodies (M-protein/paraprotein), which is essential for the diagnosis and follow-up of hematological diseases, such as multiple myeloma. Recent studies have shown that machine learning can be used to assess protein electrophoresis by, for example, examining protein glycan patterns to follow up tumor surgery. In this study we compared 26 different decision tree algorithms to identify the presence of M-proteins in human serum by using numerical data from serum protein capillary electrophoresis. For the automated detection and clustering of data, we used an anonymized data set consisting of 67,073 samples. We found five methods with superior ability to detect M-proteins: Extra Trees (ET), Random Forest (RF), Histogram Grading Boosting Regressor (HGBR), Light Gradient Boosting Method (LGBM), and Extreme Gradient Boosting (XGB). Additionally, we implemented a game theoretic approach to disclose which features in the data set that were indicative of the resulting M-protein diagnosis. The results verified the gamma globulin fraction and part of the beta globulin fraction as the most important features of the electrophoresis analysis, thereby further strengthening the reliability of our approach. Finally, we tested the algorithms for classifying the M-protein isotypes, where ET and XGB showed the best performance out of the five algorithms tested. Our results show that serum capillary electrophoresis combined with decision tree algorithms have great potential in the application of rapid and accurate identification of M-proteins. Moreover, these methods would be applicable for a variety of blood analyses, such as hemoglobinopathies, indicating a wide-range diagnostic use. However, for M-protein isotype classification, combining machine learning solutions for numerical data from capillary electrophoresis with gel electrophoresis image data would be most advantageous.

metadata and documentation made available according to the FAIR principles, and it has been assigned a permanent identifier (DOI): https://doi.org/10.5878/a2aa-kt50. To access the actual data files, a request management system is available to file a formal request to the University of Gothenburg for the data at https://snd.gu.se/en or snd@snd.gu.se. The code can be accessed through github at https://github.com/a0s6044/ProteinElectrophoresis.

**Funding:** This work is supported by the department of Clinical Chemistry, Sahlgrenska University Hospital, Gothenburg, Sweden (VRS, LMH, MN, FN, MA). There is no other specific funding for this work. The funders had no role in study design, data collection and analysis, decision to publish, or preparation of the manuscript.

**Competing interests:** The authors have declared that no competing interests exist.

**Abbreviations:** CZE, capillary zone electrophoresis; ET, Extra trees; RF, Random Forest; HGBR, Histogram Grading Boosting Regressor; LGBM, – Light Gradient Boosting Method; XGB, Extreme Gradient Boosting; CE, capillary electrophoresis; ROC, Receiver Operating Characteristic; AUC, Area under the curve; SHAP, Shapley Additive explanations; Ig, Immunoglobulin; SPEP, Serum Protein Electrophoresis; UPEP, Urine Protein Electrophoresis.

## Introduction

The analysis of fractional proteins in serum and urine (SPEP and UPEP) is included in standardized care in case of suspicion of haematological malignant chronic disease, such as multiple myeloma. Accurate diagnosis is an essential component of optimal cancer care. M-protein (also called myeloma protein or paraprotein) denotes an antibody or a fragment of an antibody that is produced in abnormal amounts by a pre-malignant or malignant plasma cell and is used as a marker for haematological malignant diseases such as myeloma as well as the asymptomatic disease monoclonal gammopathy of undetermined significance (MGUS). The size of the M-protein fraction reflects the degree of disease severity, and the level is regularly checked to follow treatment efficiency. Serum protein electrophoresis (SPEP) separates serum globulins based on their physical properties and is a diagnostic tool used to identify M-protein. In the case of malignancy, normal polyclonal immunoglobulin (Ig) production is often suppressed, and the M-protein is identified as an unusually distinct peak or atypical curve appearance within the gamma globulin fraction or beta globulin fraction, depending on the type and quantity of M-protein present.

Present-time SPEP is a semi-automated process, but involves defined laboring steps, and may be affected by intra- and inter-observer variability. The need for this type of analysis is expected to grow with an expanding population and an increased number of elderly people, which entails enhanced cancer risk, further increasing the requirement for medical expertise.

Recent studies show that machine learning can be used to assess protein electrophoresis by examining protein glycan patterns to follow up tumor surgery [1]. For example, changes in N-glycosylation patterns were analyzed using a computer-assisted machine learning method for interpreting serum proteins after surgical lung tumor resection [1]. The classification analysis resulted in a panel of N-glycans, which could be used to follow up on the effects of surgical resection of lung tumors [1]. In addition, decision tree based machine learning has been used in interpreting amino acid patterns in plasma as well as in studies based on raw signal data, such as electrocardiogram features in patients with cardiac arrhythmia, EEG signals in hearing test processes and protein mass spectra signals in serum from patients with non small cell lung cancer [2–5]. Furthermore, other types of machine learning tools have successfully been implemented in nuclear medicine and radiology for assessment of Positron emission tomography-computed tomography PET-CT or PET/CT images from 100 different organs [6,7]. Assessment using the artificial intelligence (AI)-based analysis tool Recomia showed an accuracy coefficient of 0.93 [7].

Implementing machine learning algorithms as a clinical diagnostic decision support for medical expertise in the analysis of fractionated serum proteins provides a possibility to improve quality, safety and efficiency for patients, physicians and biomedical staff and save medical resources. Thus, in this study, we investigated and compared several different decision tree algorithms for the detection of M-protein in 67,073 serum samples from 32,490 patients. Raw numerical signal data from the capillary SPEP was used and the different algorithms were evaluated for their detection accuracy as well as used for identification and verification of critical features in the data toward diagnosis.

## Materials and methods

This study was approved by the Swedish Ethical Review Authority, project ID 2021–03301, and performed in accordance with the Helsinki Declaration (as revised in 2013) and aligns with the STARD recommendations for reporting diagnostics studies. The study is a retrospective study with anonymized human clinical data with no requirement for informed consent.

The research data are by law subject to the Public Access to Information and Secrecy Act (SFS 2009:400)

## Serum protein separation

The serum samples used for this study were submitted to the laboratory for serum protein analysis in the Region Västra Götaland, Sweden, during 2015–2020. The data was accessed for research on July 5[th], 2021. All samples were analyzed on both an automated capillary electrophoresis system (Capillarys 2 Flex Piercing, SEBIA, Issy-les-Moulineaux, France), for quantification, as well as by a semi-automated agarose gel technology system (serum gel electrophoresis) (Hydrasys 2 System and PENTA PN 1260 kit, SEBIA, Issy-les-Moulineaux, France) [8,9] to not miss small fractions of M-protein. For samples where a suspected M-protein was present (based on the results from the capillary SPEP and gel SPEP), immunofixation (Hydrasys 2 System and Hydrasys IF kit, SEBIA, Issy-les-Moulineaux, France) was performed to determine the type of M-protein. M-proteins were quantified based on the capillary zone electrophoresis (CZE) (S1 Fig) using the SEBIA software (Phoresis, SEBIA, Issy-les-Moulineaux, France).

## Data processing

For the training and classification of M-proteins we used the labeled anonymized data set described above and in the Result section. All data were assessed by medical experts who determined whether M-protein was present or not based on CZE from capillary gel electrophoresis (S1 Fig) and gel pictures, and, when required, immunofixation gel pictures.

The data was randomly divided into 70% for training and 30% for testing. Using the training data, we initially taught 26 decision trees methods to detect the M-proteins. These 26 decision trees, which are part of the python sklearn package LightGBM, can quickly ascertain which methods are best for detecting M-proteins. Subsequently, we picked five of those algorithms indicated by LightGBM as performing best and trained individual python algorithms for each of those. For the individual training we randomly divided the data into 90% for training and 10% for testing. To train each of these decision trees, the algorithm minimizes the misclassification cost and specifically, in the case of boosted methods, errors of previous trees in fitting the data. During training, only the training set was used to exclusively train the models, avoiding exposure to the test set. This precaution guards against overfitting as well as data leakage, preserving the models' generalization ability. We also note that some of the models trained, such as for instance XGB, incorporate mechanisms to mitigate overfitting, ensuring robust generalization. Pruning and bagging techniques were implemented to effectively reduce the size of the resulting trees and to reduce the variance in the results of the final trees. To evaluate the performance of each tree we used metrics, such as accuracy, balanced accuracy, ROC-AUC and F1-score. The hyperparameters for all algorithms implemented in this work are set to their default values.

We installed python and relevant software libraries and programs using a docker environment with the following specifications:

Python 3.8.10, gcc 9.4.0, keras 2.6.0, open_cv 4.5.4, pyforest 1.1.0. Pandas 1.0.5 with Matplotlib 3.4.3 was used for data upload and processing. All decision trees were also based on this same version of the sklearn library. The specific parameters used for each of these trees are detailed in the Supplementary Materials and Methods section. Feature importance was based on the Shapley values from the python package shap version 0.37.0.

To evaluate the ability of the algorithm to determine the M-protein isotypes, we used precision scores and recall scores, as well as confusion matrix analysis.

## Results

For this study we used a total of 67,073 patient serum samples from 32,940 individuals of which 15,684 samples were positive for M-protein. The average age was 57±21.5 years. The gender distribution was 15,164 men and 17,776 women. The patient demographics and distribution of monoclonal isotypes are shown in Fig 1, Tables 1 and 2 and S2 Fig.

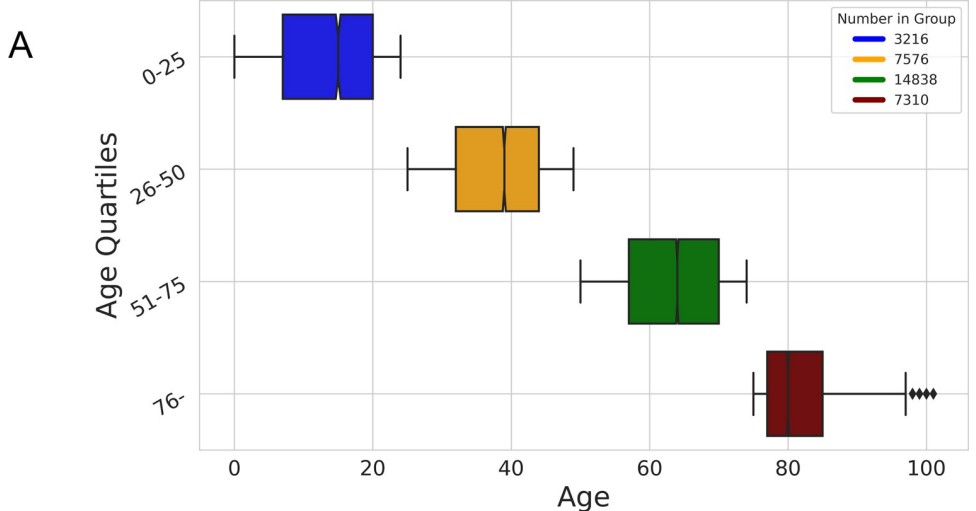

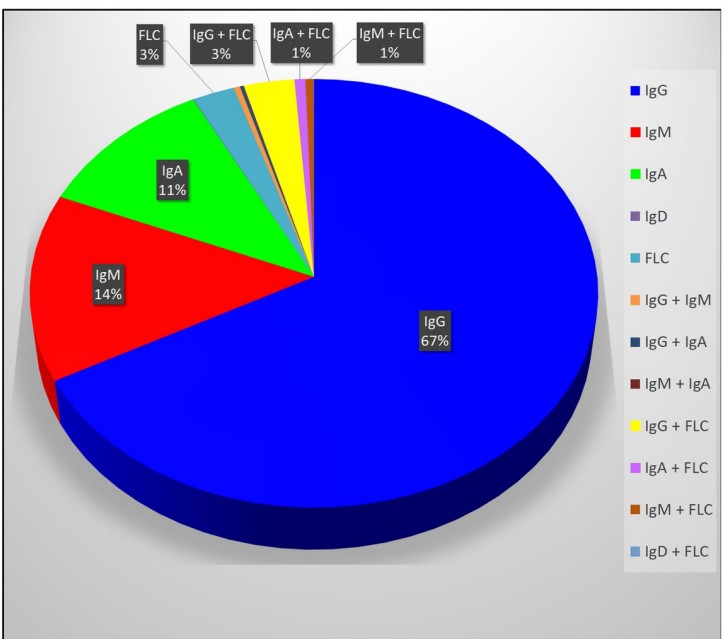

**Fig 1. Patient demographics and M-protein isotype distribution.** *(A),* Age distribution and number of individuals within age quartiles for the individuals included in the study, total n = 32,940. Age is shown as mean ± standard deviation. *(B),* M-protein isotype distribution among the M-protein positive samples included in the study, n = 15,684.

**Table 1. M-protein isotype distribution among the M-protein positive samples included in the study.**

| M-protein isotype | Number of M-protein positive samples | % |
|---|---|---|
| IgG | 10,538 | 67.2 |
| IgM | 2,215 | 14.1 |
| IgA | 1,745 | 11.1 |
| IgD | 9 | 0.06 |
| FLC | 404 | 2.6 |
| IgG + IgM | 60 | 0.4 |
| IgG + IgA | 33 | 0.2 |
| IgM + IgA | 6 | 0.04 |
| IgG + IgD | 1 | 0.01 |
| IgG + FLC | 488 | 3.1 |
| IgA + FLC | 102 | 0.7 |
| IgM + FLC | 81 | 0.5 |
| IgD + FLC | 2 | 0.01 |
| Total | 15,684 | |

## Decision tree methods detect M-proteins with high accuracy

In order to compare the accuracy of different decision tree-based methods to detect the existence of M-proteins in the serum samples, we evaluated 26 methods by training them for identification of M-proteins on 50,391 samples and then testing them on 16,797 unseen samples. The methods showing the highest accuracy were Extra Trees (ET), Random Forest (RF), Histogram Grading Boosting Regressor (HGBR), Light Gradient Boosting Method (LGBM), and Extreme Gradient Boosting (XGB), resulting in accuracy scores between 0.962 and 0.977 (Table 3). Further classification testing using receiver operating characteristic (ROC) area under the curve (AUC) calculations and F1 score calculations confirmed superior ability of these methods to classify unseen data (Table 3 and Fig 2). The ROC AUC scores reached as high as 0.993 (Table 3 and Fig 2) and the F1 scores reached as high as 0.977 (Table 3). Accuracy scores, ROC AUC values and F1 scores for the remaining methods tested are listed in S1 Table. The computational cost of training all the algorithms was in total less than 20 minutes in a CPU Intel i7-9700E 8-core processor. Once trained each algorithm was able to produce results in real time.

## M-protein detection by decision tree methods can be verified to the gamma and beta fractions

The majority of machine learning algorithms are unable to communicate how they produce a particular outcome, a phenomenon known as the "black box" [10]. However, this knowledge is important for our purposes in terms of understanding how the input data leads to the M-protein diagnosis as well as verifying that the algorithm is processing the input data correctly.

**Table 2. Distribution of concentrations for IgG, IgA and IgM M-proteins, divided into three groups: <1 g/l, 1–5 g/l and >5 g/l.**

| M-protein concentration | <1 g/l | 1–5 g/l | >5 g/l |
|---|---|---|---|
| Number of IgG M-protein samples | 16 | 2,822 | 7,700 |
| Number of IgA M-protein samples | 3 | 396 | 1,346 |
| Number of IgM M-protein samples | 3 | 663 | 1,549 |
| Total IgG+IgA+IgM M-protein samples | 22 | 3,881 | 10,595 |

**Table 3. The top five decision tree methods for identification of M-protein.**

| Classifier | Accuracy | Balanced Accuracy | ROC AUC | F1 Score |
|---|---|---|---|---|
| ET | 0.977 | 0.950 | 0.993 | 0.977 |
| RF | 0.973 | 0.942 | 0.989 | 0.973 |
| HGBR | 0.962 | 0.942 | 0.988 | 0.961 |
| LGBM | 0.962 | 0.902 | 0.987 | 0.961 |
| XGB | 0.965 | 0.912 | 0.985 | 0.964 |

ET = Extra Trees, RF = Random Forest, HGBR = Histogram Grading Boosting Regressor, Light LGBM = Gradient Boosting Method, and XGB = Extreme Gradient Boosting.

We aimed to resolve this issue using Shapley Additive explanations (SHAP) [11,12]. This method allows us to understand which of the input data features are most important in order to discriminate between samples containing M-proteins and samples not containing M-proteins. Practically, this task is equivalent to optimizing the classification outcome of the algorithm under all possible combinations of input data features. A successful outcome was one which, for our application, correctly predicted the presence of M-protein. The CZE input data was divided into 300 features, each corresponding to the migration time on the x-axis of the CZE (Fig 3A). The results verified the gamma globulin fraction and part of the beta-2 globulin fraction as the most important for detecting M-protein (Figs 3A and S3), corresponding to the features with migration times between 240 and 262 seconds in the CZE (Fig 3B), thereby verifying the results and further strengthening the reliability of the top five algorithms. Furthermore, using SHAP values also enabled us to compute an M-protein probability score for each patient (Fig 3C), which can be used to predict the likelihood of M-protein being present for a

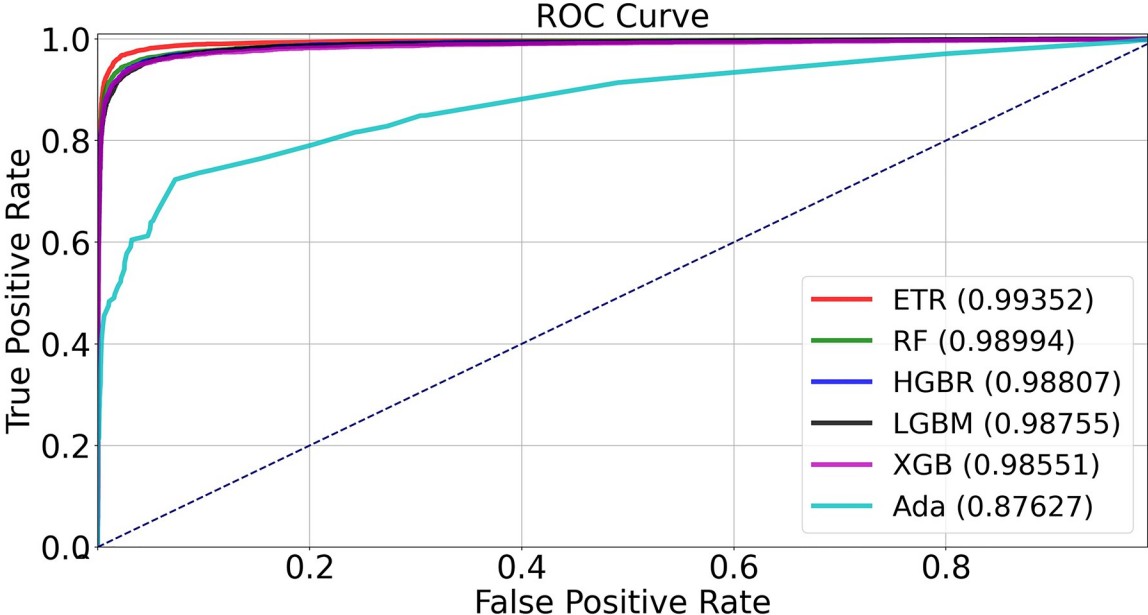

**Fig 2. Receiver operating characteristic (ROC) calculations.** ROC curves for the five most successful decision tree algorithms: Extra Trees (ET), Random Forest (RF), Histogram Grading Boosting Regressor (HGBR), Light Gradient Boosting Method (LGBM), and Extreme Gradient Boosting (XGB). A less successful algorithm (ADA) is included as a reference. The inset shows the corresponding area under the curve (AUC) values.

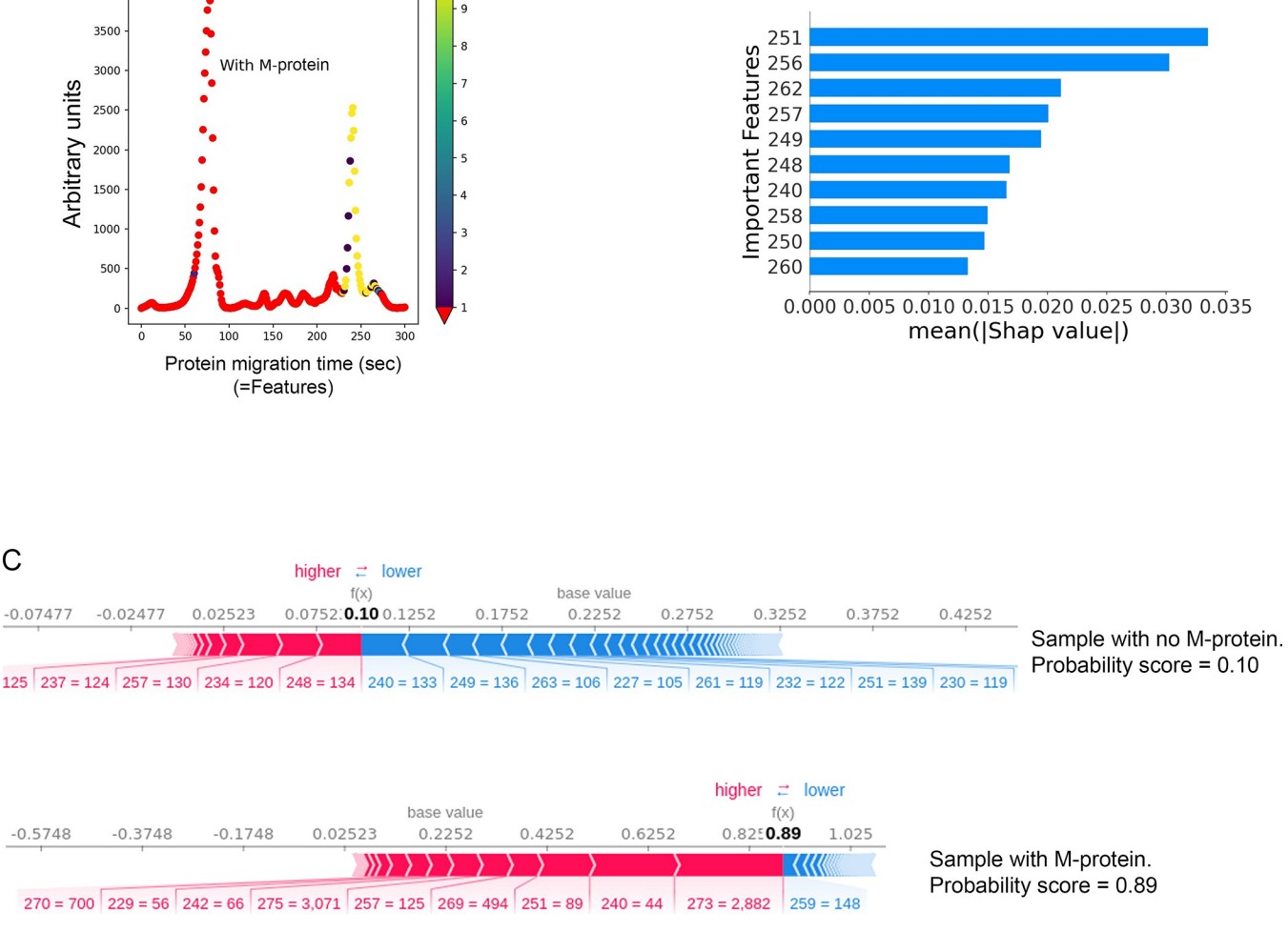

**Fig 3. Calculations for Shapley Additive explanations (SHAP).** *(A)*, Capillary zone electrophoresis (CZE) from a sample containing M-protein. The x-axis is divided into 300 time points (features), representing the migration time during SPEP. The most important features for detecting M-protein based on the SHAP calculations are highlighted in yellow, clearly verifying the gamma globulin fraction and part of the beta-2 globulin fraction as the most important for detecting M-protein. *(B)*, The ten most important features (from a total of 300) for detection of M-protein. Each feature value corresponds to the migration time on the x-axis of the CZE. The x-axis shows the mean SHAP value. *(C)*, SHAP values for a sample with no M-protein (top) and with M-protein (bottom). Red numbers indicate positively correlated features for detecting M-protein and blue numbers indicate negatively correlated features. Probability scores for the presence of M-protein in the samples are shown in bold numbers. Features with their corresponding SHAP values are indicated below each panel.

specific patient. For comparison, we also performed the same analysis with a classic feature importance approach, which gave limited results (S4 Fig). Whereas the classic approach primarily identifies features corresponding to the gamma fraction, SHAP identifies features for both the beta and gamma fractions (Figs 3C and S4).

## Extra trees and random forest show the best success rate in classifying M-protein isotypes

We next wanted to test the ability of the top five scoring methods in the initial analysis above (ET, RF, HGBR, LGBM, and XGB) to correctly determine the isotype of the M-protein present in a sample. The scoring for small M-spikes was poorer than larger ones as expected (data not shown), but to what degree is impossible to evaluate with the low number of samples we have in the small size category in our data set. For this purpose, we had to exclude the samples with

**Table 4. Ability of the five best algorithms to determine the M-protein isotypes.**

| Extra trees | Total true number in the testing population | Precision score | Recall score | F1 score |
|---|---|---|---|---|
| Healthy | 5,300 (79.1%) | 0.96 | 0.99 | 0.98 |
| IgG isotype | 1,020 (15.2%) | 0.84 | 0.9 | 0.87 |
| IgA isotype | 185 (2.7%) | 0.95 | 0.29 | 0.44 |
| IgM isotype | 199 (3.0%) | 0.95 | 0.2 | 0.32 |
| **Random forest** | **Total true number in the testing population** | **Precision score** | **Recall score** | **F1 score** |
| Healthy | 5,300 (79.1%) | 0.95 | 0.99 | 0.97 |
| IgG isotype | 1,020 (15.2%) | 0.81 | 0.87 | 0.84 |
| IgA isotype | 185 (2.7%) | 0.98 | 0.25 | 0.40 |
| IgM isotype | 199 (3.0%) | 0.91 | 0.15 | 0.25 |
| **HGBR** | **Total true number in the testing population** | **Precision score** | **Recall score** | **F1 score** |
| Healthy | 5,300 (79.1%) | 0.93 | 1.00 | 0.96 |
| IgG isotype | 1,020 (15.2%) | 0.81 | 0.75 | 0.78 |
| IgA isotype | 185 (2.7%) | 0.97 | 0.21 | 0.34 |
| IgM isotype | 199 (3.0%) | 0.76 | 0.08 | 0.15 |
| **LGBM** | **Total true number in the testing population** | **Precision score** | **Recall score** | **F1 score** |
| Healthy | 5,300 (79.1%) | 0.93 | 1.00 | 0.96 |
| IgG isotype | 1,020 (15.2%) | 0.81 | 0.76 | 0.79 |
| IgA isotype | 185 (2.7%) | 0.97 | 0.21 | 0.34 |
| IgM isotype | 199 (3.0%) | 0.84 | 0.08 | 0.15 |
| **XGB** | **Total true number in the testing population** | **Precision score** | **Recall score** | **F1 score** |
| Healthy | 5,300 (79.1%) | 0.94 | 1.00 | 0.97 |
| IgG isotype | 1,020 (15.2%) | 0.83 | 0.77 | 0.80 |
| IgA isotype | 185 (2.7%) | 0.95 | 0.29 | 0.44 |
| IgM isotype | 199 (3.0%) | 0.62 | 0.24 | 0.35 |

Total number of samples in the test population = 6,704.

Precision score: Proportion of correctly predicted M-protein isotypes in the test sample population.

Recall score: Proportion of correctly predicted M-protein isotypes out of the true isotype populations.

F1 score = the harmonic mean of the precision and the recall scores.

free light chain M-protein, IgD M-protein, and samples with more than one type of M-protein in our data set, since these samples were too few to be properly trained and tested by the algorithms. Out of the 6,704 samples used for testing, the ET algorithm was most successful in scoring IgG M-proteins, with an F1 score of 0.87 (Table 4). The Random forest and XGB algorithms reached an F1 score of 0.44 for IgA M-proteins (Table 4), whereas XGB reached an F1 score of 0.35 for IgM proteins.

To evaluate the isotype classification performance of the algorithms in more detail, a confusion matrix analysis was performed for each of the five algorithms (Table 5). This analysis verified ET and XGB as the most successful algorithms in scoring IgG, IgA and IgM, and, in addition, showed that ET had fewer false negative scores compared to the other four algorithms. Taken together, out of the 6,704 samples in the test population, ET and XGB showed a higher classification success rate compared to RF, HGBR, and LGBM.

## Discussion

SPEP is a standard screening method for evaluating immunoglobulin patterns. A specific group of diseases, gammopathies, gives rise to an atypical peak in the histogram gamma

**Table 5. Confusion matrices showing the isotype classification performance for the five best algorithms.**

| Extra trees | Healthy | IgG isotype | IgA isotype | IgM isotype |
|---|---|---|---|---|
| Healthy | 5273 | 25 | 2 | 0 |
| IgG isotype | 101 | 916 | 1 | 2 |
| IgA isotype | 111 | 21 | 53 | 0 |
| IgM isotype | 27 | 133 | 0 | 39 |
| Proportion false positives | 4% | 5% | 5% | 16% |
| **Random forest** | **Healthy** | **IgG isotype** | **IgA isotype** | **IgM isotype** |
| Healthy | 5261 | 38 | 1 | 0 |
| IgG isotype | 132 | 885 | 0 | 3 |
| IgA isotype | 108 | 31 | 46 | 0 |
| IgM isotype | 29 | 141 | 0 | 29 |
| Proportion false positives | 5% | 2% | 9% | 19% |
| **HGBR** | **Healthy** | **IgG isotype** | **IgA isotype** | **IgM isotype** |
| Healthy | 5280 | 20 | 0 | 0 |
| IgG isotype | 245 | 770 | 0 | 5 |
| IgA isotype | 124 | 23 | 38 | 0 |
| IgM isotype | 44 | 138 | 1 | 16 |
| Proportion false positives | 7% | 3% | 24% | 19% |
| **LGBM** | **Healthy** | **IgG isotype** | **IgA isotype** | **IgM isotype** |
| Healthy | 5277 | 23 | 0 | 0 |
| IgG isotype | 238 | 778 | 1 | 3 |
| IgA isotype | 123 | 24 | 38 | 0 |
| IgM isotype | 51 | 132 | 0 | 16 |
| Proportion false positives | 7% | 3% | 16% | 19% |
| **XGB** | **Healthy** | **IgG isotype** | **IgA isotype** | **IgM isotype** |
| Healthy | 5275 | 22 | 2 | 1 |
| IgG isotype | 211 | 781 | 0 | 28 |
| IgA isotype | 102 | 30 | 53 | 0 |
| IgM isotype | 43 | 107 | 1 | 48 |
| Proportion false positives | 6% | 5% | 38% | 17% |

Total number of samples in the test population = 6,704.

Rows show actual classification numbers.

Columns show predicted isotype values.

Boxes highlighted in green represent true positives.

Remaining values for each column represent false positives.

Remaining values for each row represent false negatives.

Example: Out of a total of 6,704 samples, 5,273 samples were correctly classified as healthy by the Extra tree algorithm. 101 samples were falsely classified as healthy, but were positive for IgG M-protein. 2 healthy samples were falsely classified as IgA M-protein.

globulin or beta globulin regions following serum electrophoresis due to the presence of M-protein (or paraprotein), caused by abnormal proliferation of a single clone of plasma cells.

Recent reports have shown that machine learning can be a useful decision support tool for various clinical analyses including examining protein glycan patterns to follow up the effects of surgical resection of lung tumors [1], interpreting amino acid patterns in plasma as well as in studies using raw signal data rather than data related to concentration of specific molecules [3–5]. Thus, in the current study, we aimed to investigate and compare several different machine learning algorithms for M-protein identification in human serum samples following SPEP.

Using 26 different decision tree algorithms we trained and tested them on our data set, which included 67,073 patient serum samples, to evaluate which methods were superior in accurately determining the presence of M-protein. Five methods, Extra Trees, Random Forest, Histogram Grading Boosting Regressor, Light Gradient Boosting Method, and Extreme Gradient Boosting, were found to generate highly accurate identification scores, ranging from 96% to 98% accuracy and ROC AUCs of 0.99 for all methods. Chabrun et al recently reported an AI application for SPEP analysis, based on four deep-learning models that included numerous parameters [13]. Their AI application generated M-protein classification accuracy scores of 91.2% - 93.1% although the ROC AUC reached as high as 0.99. The discrepancy in accuracy score between our model and the model by Chabrun et al is likely most attributable to less precise annotations due to difficulty in clearly defining M-spikes in some SPEP samples, a common problem when interpreting conditions with restricted heterogeneity in immunoglobulin fractions. Our data was based on samples analyzed by multiple assays, which allowed robust results and thereby more clearly detected and defined M-spikes.

Complex machine learning algorithms, including deep learning models, as well as methods like XGBoost and LightGBM, share a common limitation referred to as the 'black box' phenomenon, where understanding the internal decision-making process leading to outcomes becomes challenging [10]. This often leads to uncertainty regarding the reliability of the method used. In addition, the "black box" phenomenon prevents from identifying novel patterns and properties within the data set that could contribute to better understand and interpret underlying features of a disease. Explanation methods, such as Shapley Additive explanations (SHAP), can be employed to enhance the interpretability of these models by providing insights into feature importance and contributing factors. Other groups have tried to combine deep learning algorithms with decision trees [14,15] or applied natural language processing approaches [16] to classify and predict multiple myeloma or other diseases, all with accuracy scores of <93%. Combining deep neural networks and decision trees requires heavy processing of input data and substantial tuning of model parameters which can explain the lower predictor accuracy and F1 scores making these types of models difficult to apply in practice.

While decision tree methods can offer increased transparency due to their rule-based nature, it is important to note that this transparency might diminish as we move to more complex algorithms like ensemble models, including XGBoost (XGB) and LightGBM (LGBM). These ensemble methods can introduce additional complexity that could impact the interpretability of the models and can make it harder to identify suitable values of their hyperparameters. By utilizing decision tree algorithms based on the raw numerical data generated from capillary gel electrophoresis, we were able to elucidate the features responsible for the detection of M-proteins using Shapley Additive explanations (SHAP) [11,12], thereby verifying the gamma globulin fraction and part of the beta-2 globulin fraction as the M-protein hallmarks for the algorithm outcome (M-protein of IgA type is generally found in the beta-2 zone). This is a crucial element of our approach, not only for verifying already known patterns and features, but particularly for identifying possible novel patterns when using the method for other applications. Elucidating previously unidentified patterns and features in a dataset could largely improve the understanding and interpretation of the analysis results, thereby increasing the sensitivity and quality of the analysis. Ultimately, this leads to better care for the patient.

Taking advantage of machine learning methods as a support tool for the evaluation and determination of blood proteins creates possibilities to not only increase the accuracy of the results and safety for the patient, but possibly also extensive savings of resources. By using machine learning methods to eliminate negative samples (e.g., samples with no M-protein

present), the medical and biomedical staff can focus on the positive samples (e.g., determining the type of M-protein present in the sample). Implementing SHAP calculations further adds value in this context by facilitating predictions and introduction of appropriate cut off margins. Furthermore, using machine learning methods as a support tool can help reduce the amount of unnecessary additional analyses which are often performed due to uncertainty and/ or inexperience of the person analyzing the initial electrophoresis result.

A limitation with our approach was the low accuracy scores for identifying the specific isotype of M-proteins present. In the context of identifying specific M-protein isotypes, it is important to consider the potential impact of class imbalance within the data set. The relatively low accuracy scores observed in our approach may be attributed to the presence of certain M-protein isotypes that are inherently less prevalent within the population. This imbalance in class distribution can pose challenges for accurate classification. Decision tree methods are generally effective in tackling class imbalance within the data set [17] by their inherent ability to prioritize minority classes [18], thanks to impurity-based splitting criteria like the Gini index or entropy or ensemble methods [19] like Random Forests or gradient boosting. However, to mitigate this issue and enhance the model's ability to discern rarer isotypes, techniques such as SMOTE (Synthetic Minority Over-sampling Technique) could be explored. By generating synthetic instances of underrepresented isotypes, SMOTE aims to balance class proportions and improve model performance. The incorporation of such methods could help to further address the impact of class imbalance and potentially lead to more accurate identification of specific M-protein isotypes.

Other likely contributing factors for the low accuracy scores for identifying specific M-protein isotypes is that samples with small M-protein fractions often are difficult to distinguish in the CZE. Furthermore, abnormal patterns other than those resulting from the presence of M-proteins could confound the results generated by a machine learning tool. Such abnormal patterns could be identified in future work using anomaly detection methods. For a more accurate scoring of M-protein isotypes, a combination of decision tree methods and deep learning algorithms of immunofixation electrophoresis, such as that recently described by Hu et al [20], would likely be more advantageous.

The incidence of samples with free light chains in our data set is low compared to what is generally observed and reported in the literature. Light chain multiple myeloma (LCMM) accounts for approximately 15% of all cases of multiple myeloma [21] and free light chain M-proteins are often observed in small quantities in serum when analyzed by SPEP. Since M-proteins of very small concentrations are difficult to detect in the CZE, it is important to take into account that this may have influenced our results.

In conclusion, we found that decision tree machine learning algorithms can identify the presence of M-proteins in human serum following SPEP with high accuracy, using routinely collected laboratory data. In addition, we were able to verify the gamma globulin and beta-2 globulin fractions as the hallmark for the algorithm outcome, further enhancing the reliability of the method. The use of machine learning algorithms for the prediction of overall prognosis is a promising new area that can support doctors with clinical decisions, improve diagnosis accuracy and safety for the patients as well as markedly reduce the volume of resources needed.

## Supporting information

**S1 Fig. Capillary zone electrophoresis (CZE) outlining the different serum protein fractions in a normal sample and in a sample containing M-protein.**
(TIF)

**S2 Fig. Concentration range of free light chains among the samples in the data set.**
(TIF)

**S3 Fig. SHAP calculation results from nine samples containing M-protein and three samples with no M-protein.**
(TIF)

**S4 Fig. Classic feature importance approach based on the trained Extra Trees Classifier.**
(TIF)

**S1 Table. Accuracy scores, ROC AUC values and F1 scores for the remaining methods tested.**
(TIF)

## Author Contributions

**Conceptualization:** Maria Nilsson, Lillemor Mattsson Hultén, Victoria Rotter Sopasakis.

**Data curation:** Alexandros Sopasakis.

**Formal analysis:** Alexandros Sopasakis.

**Methodology:** Alexandros Sopasakis.

**Project administration:** Lillemor Mattsson Hultén, Victoria Rotter Sopasakis.

**Resources:** Maria Nilsson, Mattias Askenmo, Fredrik Nyholm, Victoria Rotter Sopasakis.

**Software:** Alexandros Sopasakis.

**Supervision:** Lillemor Mattsson Hultén, Victoria Rotter Sopasakis.

**Writing – original draft:** Alexandros Sopasakis, Victoria Rotter Sopasakis.

**Writing – review & editing:** Alexandros Sopasakis, Maria Nilsson, Mattias Askenmo, Fredrik Nyholm, Lillemor Mattsson Hultén.

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
