## [Decision Letter · Decision Letter 0]

10 Aug 2023

PONE-D-23-17568Machine learning evaluation for identification of M-proteins in human serumPLOS ONE

Dear Dr. Rotter Sopasakis,

Thank you for submitting your manuscript to PLOS ONE. After careful consideration, we feel that it has merit but does not fully meet PLOS ONE’s publication criteria as it currently stands. Therefore, we invite you to submit a revised version of the manuscript that addresses the points raised during the review process.

We look forward to receiving your revised manuscript.

Kind regards,

John Adeoye

Academic Editor

PLOS ONE

"This work is supported by the department of Clinical Chemistry, Sahlgrenska University Hospital, Gothenburg, Sweden (VRS, LMH, MN, FN, MA). There is no other specific funding for this work."         

6. We note that you have indicated that data from this study are available upon request. PLOS only allows data to be available upon request if there are legal or ethical restrictions on sharing data publicly. For more information on unacceptable data access restrictions, please see http://journals.plos.org/plosone/s/data-availability#loc-unacceptable-data-access-restrictions.

7. We note that you have stated that you will provide repository information for your data at acceptance. Should your manuscript be accepted for publication, we will hold it until you provide the relevant accession numbers or DOIs necessary to access your data. If you wish to make changes to your Data Availability statement, please describe these changes in your cover letter and we will update your Data Availability statement to reflect the information you provide.

8. Please include captions for your Supporting Information files at the end of your manuscript, and update any in-text citations to match accordingly. Please see our Supporting Information guidelines for more information: http://journals.plos.org/plosone/s/supporting-information

Additional Editor Comments:

The reviewers both agree that the dataset used for ML model construction is valuable but both cited the poor methodological approach of the study. In view of this, the authors are advised to revise their ML modelling methodology extensively in line with contemporary standards to allow for further consideration of their manuscript in the Journal.

Reviewers' comments:

Reviewer's Responses to Questions

**Comments to the Author**

1. Is the manuscript technically sound, and do the data support the conclusions?

Reviewer #1: Yes

Reviewer #2: Partly

2. Has the statistical analysis been performed appropriately and rigorously? 

Reviewer #1: Yes

Reviewer #2: No

3. Have the authors made all data underlying the findings in their manuscript fully available?

Reviewer #1: Yes

Reviewer #2: Yes

4. Is the manuscript presented in an intelligible fashion and written in standard English?

Reviewer #1: Yes

Reviewer #2: Yes

5. Review Comments to the Author

Reviewer #1: Please see the attached file for my comments, in addition with the summary of my review below:

The authors performed capillary & gel electrophoresis and immunofixation for more to 60k serum samples. Based on their data, they classified their samples into either “absence of M-protein” or “presence of M-protein”, and classified the subtype of M-protein, when relevant. Next, they trained various ML classifiers to automate those classification tasks, by feeding those with raw numerical data of the SPEP (Serum Protein Electrophoresis) curves, i.e. the value of each point of the SPEP curve.

The authors present an extremely valuable dataset: more than 60k samples, annotated thanks to three biological assays: capillary SPEP, gel SPEP and immunofixation; with the interpretation from laboratory experts. This alone is a solid argument supporting the publication of their results. It should be noted that the data might be described better, for instance by 1) indicating the size of M-spikes in their curves and the proportion of small/medium/large M-spikes, for instance; and 2) the proportion of other abnormal patterns which may mask or be confounded with M-proteins (artifacts (fibrinogen, iodinated contrast agents…), beta-gamma bridging). Furthermore, it is not clear if all samples were analyzed with all 3 assays: capillary SPEP, gel SPEP and immunofixation. A table may be useful here. Finally, the choice of the authors to partition their data into a training and test sets only, without any validation set, despite their number of samples, is at major risk of data leakage: there is an urge in addressing or at least discussing this point in the manuscript.

The main downside of this study, in my opinion, is the machine learning methodology used here, which is far from state of the art standards. SPEP data, is, by definition, signal data. Multiple works have highlighted the superiority of DL (deep learning) methods, mostly but not limited to CNN & transformers, to such data. However, the word “signal” is never used throughout the manuscript and DL models are absent from the models trained by the authors. The question of why the authors chose to elude this major point stays unanswered at the end of this manuscript. The ML (meachine learning) knowledge of the authors does not seem to be an obstacle to the use of DL models, based on the expertise they demonstrate in ML and DL, largely discussing the upsides and downsides of various ML models including DL ones. Furthermore, training/inference speed should not be an issue either, since 60k samples times 300 points per curve is a relatively small amount of data to process (the authors state that 20 minutes were needed to train all 26 models in this study).

The use of SHAP values, here, further highlights this problem. The models seem to simply “look at high values” in the beta & gamma regions and deduce if there is an M-spike in the sample. However, M-spikes are not always characterized by high values, and small M-spikes may only be visually detected, by the expert, thanks to an abnormal qualitative pattern, rather than by observing high quantities in the beta/gamma fractions (e.g., shoulder in the beta fraction). This is only possible with ML if treating the data as a signal, e.g., by using convolutional layers. Since the authors do not inform about the exact patterns observed in their dataset (see my previous paragraph about data), it is impossible to predict their models’ expected behavior on such samples. Unfortunately, those samples are crucial, and highly responsible for the fact that SPEP is not yet fully automated in modern laboratories.

Finally, the authors imply that their methodology may have other applications. This is highly doubtful, as they are few examples of biological assays outputting highly standardized/aligned signal data, which may give such robust results with the methodology they use. Indeed, the previous studies the authors cite to support their work have been using this kind of ML tools to make predictions based on concentrations deducted from signal data, rather than the raw data themselves.

In total, this study seems highly promising, thanks notably to a highly valuable dataset, but which should be analyzed using state of the art machine learning tools to be considered in today’s literature.

Reviewer #2: This is a very interesting paper with a clear clinical question ( The major clinical question is the presence of monoclonal fraction(s) of antibodies (M-protein/paraprotein), which is essential for the diagnosis and follow-up of hematological diseases, such as multiple myeloma) . They also have a very nice dataset.

Regarding the adopted evaluation procedures, there is an important step that is missing: how did the authors choose the hyperparameters? Usually , one performs a gridsearch using k-fold in the training set as GridSeachCV form sk-learn does. If you use the test set to determine the best set of hyperparameters, the results woulb be biased.

it would be nice to compare the results provided by SHAP with standard feature importance that several tree-based algorithms can provide.

In the discussion , the authors say “Deep learning algorithms are highly suitable and powerful for image analysis but exhibit some disadvantages. One such weakness is the “black box” phenomenon, the inability to explain how the outcome result was achieved”. One can say the same thing about XGB or LGBM. For deep learning , there are several explanation methods that could be employed, including SHAP. Please correct this statement.

In the discussion the authors also say “The use of decision tree methods can be more transparent compared to deep learning algorithms, if implemented as we propose here, but work best for numerical series of data rather than data retrieved from images”. Decision trees are more transparent, but this is not true for the other algorithms derived from them as XGB. It is also very hard to tune XGB or LGBM hyperparameters, because there are several of them(see the documentation)

The authors say that the limitation of their approach was the relatively low accuracy

scores for identifying the specific isotype of M-protein present, probably due to the low number of certain M-protein isotypes in the population. What the authors had was an imbalanced dataset. This could be correct using SMOTE techniques.

6. PLOS authors have the option to publish the peer review history of their article (what does this mean?). If published, this will include your full peer review and any attached files.

Reviewer #1: **Yes: **Floris CHABRUN

Reviewer #2: No

---

## [Author Response · Author response to Decision Letter 0]

16 Nov 2023

We thank the Reviewers for their constructive and insightful comments and suggestions, which we believe have considerably improved our manuscript. We have fully addressed the reviewers’ comments and concerns in a point-by-point response, submitted as requested as a "Response to Reviewers" file. All changes made to the manuscript are shown with track changes in the “Revised Article with Changes Highlighted” file. We hope that the reviewing process finds our revised manuscript acceptable for publication in PLOS ONE.

---

## [Decision Letter · Decision Letter 1]

9 Feb 2024

PONE-D-23-17568R1Machine learning evaluation for identification of M-proteins in human serumPLOS ONE

Dear Dr. Rotter Sopasakis,

Thank you for submitting your manuscript to PLOS ONE. After careful consideration, we feel that it has merit but does not fully meet PLOS ONE’s publication criteria as it currently stands. Therefore, we invite you to submit a revised version of the manuscript that addresses the points raised during the review process.

We look forward to receiving your revised manuscript.

Kind regards,

John Adeoye

Academic Editor

PLOS ONE

Journal Requirements:

Reviewers' comments:

Reviewer's Responses to Questions

**Comments to the Author**

1. If the authors have adequately addressed your comments raised in a previous round of review and you feel that this manuscript is now acceptable for publication, you may indicate that here to bypass the “Comments to the Author” section, enter your conflict of interest statement in the “Confidential to Editor” section, and submit your "Accept" recommendation.

Reviewer #1: (No Response)

Reviewer #3: All comments have been addressed

2. Is the manuscript technically sound, and do the data support the conclusions?

Reviewer #1: Yes

Reviewer #3: Yes

3. Has the statistical analysis been performed appropriately and rigorously? 

Reviewer #1: Yes

Reviewer #3: Yes

4. Have the authors made all data underlying the findings in their manuscript fully available?

Reviewer #1: Yes

Reviewer #3: Yes

5. Is the manuscript presented in an intelligible fashion and written in standard English?

Reviewer #1: Yes

Reviewer #3: Yes

6. Review Comments to the Author

Reviewer #1: First, I want to thank the authors for carefully studying and replying to each and every one of my comments, and for the thorough work I believed they have undertaken for improving their manuscript. In my opinion, all major concerns regarding this study have been addressed.

Particularly, the current version of their manuscript now enables a detailed understanding of the exact data used by the authors and potentially accessible to the community, notably due to the addition of Tables 1 and 2 and confirmation that all samples were double checked with both CZE and gel SPEP. This confirms the high value of those data.

I still have a few minor comments to the authors, which may or may not trigger modifications in the final manuscript, to their discretion:

- New tables 1 and 2 are interesting and I feel are nice additions to the manuscript. In Table 2, I am not sure about the use of the word “tertiles”, which implies that selected cut-offs, namely 1g/L and 5g/L, would divide the dataset into three subsets of equal size (1 third of the samples), which is not the case here.

- Several arguments against the use of deep learning in this study seem fallacious in my opinion. For instance:

o the assumed ability of tree models to better handle class imbalance due to Gini index/entropy overlooks the fact that cross entropy is one of the most widely used loss functions in deep learning; for the very same reason.

o the authors imply that convolutional layers would not be suitable to the analysis of 306-point-wide traces reshaped to 17x18 images. Analyzing small images is perfectly performed by CNN, for instance on countless MNIST examples online (28x28 images). Furthermore, 306-wide traces can (and should here) be reshaped to (306,1) arrays/tensors to avoid jeopardizing spatial information, as described in several previous works analyzing 1-dimensional signal. Finally, authors could even use 1d-conv layers to avoid reshaping raw traces if reshaping itself is a concern.

o I can understand that the authors refuse to use DL compared to tree models, and I think the “no free lunch theorem” sufficiently justifies this strategy. But in my opinion, the specific arguments cited above are misleading and should be removed from the final version.

- Authors corrected the name of the python package they used from “shapely” to “Shapely” version 1.7.1. I would like to make sure this is not a mistake: Figure 3 highly resembles plots obtained with the “SHAP” package based on “ShapLEy” (not “ShapELy”) values. Furthermore, SHAP is also cited multiple times by the author. ShapELy v 1.7.1 seems to be a Python package related to geometric analysis rather than feature importance. Can the authors confirm there is no typo here?

- Knowing that the authors, even unknowingly, complied with recommendations such as STARD is extremely positive. I think the readership’s trust would highly benefit from citing this in the Methods section, though this is not mandatory.

- The table cited in the “Response to reviewers” file, depicting the performance of models according to the M-spike concentration (<1g/L, 1-5g/L, >5g/L) is really interesting, and in my opinion its addition in the manuscript or supplementary material, or at least a sentence in the Results would be of interest for the readers of this work.

Reviewer #3: This is an interesting study that demonstrates the analysis of a large set of electrophoresis data using machine learning for the identification of M-proteins in serum. Two referees have thoroughly assessed the paper and provided useful feedback which has been carefully addressed by the authors. While I agree that the ML algorithms used are not overly sophisticated, their applicability to the problem makes the procedure easy to implement. It would be good if this could be tested on 'new' electrophoresis data from various sources to test the transferability of the method, but I am aware this is beyond the scope of the current study. The dataset is extensive and relevant and I think the paper will make a good addition to the literature.

7. PLOS authors have the option to publish the peer review history of their article (what does this mean?). If published, this will include your full peer review and any attached files.

Reviewer #1: **Yes: **Floris Chabrun

Reviewer #3: No

---

## [Author Response · Author response to Decision Letter 1]

10 Feb 2024

Reviewer #1: First, I want to thank the authors for carefully studying and replying to each and every one of my comments, and for the thorough work I believed they have undertaken for improving their manuscript. In my opinion, all major concerns regarding this study have been addressed.

Particularly, the current version of their manuscript now enables a detailed understanding of the exact data used by the authors and potentially accessible to the community, notably due to the addition of Tables 1 and 2 and confirmation that all samples were double checked with both CZE and gel SPEP. This confirms the high value of those data.

We thank the reviewer for carefully and thoroughly assessing our manuscript and our changes and responses in the first round of revision. We have addressed the new comments as seen below in a point-by-point response. We hope that our responses have resolved any remaining uncertainties that the reviewer had.

I still have a few minor comments to the authors, which may or may not trigger modifications in the final manuscript, to their discretion:

- New tables 1 and 2 are interesting and I feel are nice additions to the manuscript. In Table 2, I am not sure about the use of the word “tertiles”, which implies that selected cut-offs, namely 1g/L and 5g/L, would divide the dataset into three subsets of equal size (1 third of the samples), which is not the case here.

We have removed the word “tertile” from Table 2 as requested and instead added “three groups: <1 g/l, 1-5 g/l and >5 g/l”.

- Several arguments against the use of deep learning in this study seem fallacious in my opinion. For instance:

o the assumed ability of tree models to better handle class imbalance due to Gini index/entropy overlooks the fact that cross entropy is one of the most widely used loss functions in deep learning; for the very same reason.

o the authors imply that convolutional layers would not be suitable to the analysis of 306-point-wide traces reshaped to 17x18 images. Analyzing small images is perfectly performed by CNN, for instance on countless MNIST examples online (28x28 images). Furthermore, 306-wide traces can (and should here) be reshaped to (306,1) arrays/tensors to avoid jeopardizing spatial information, as described in several previous works analyzing 1-dimensional signal. Finally, authors could even use 1d-conv layers to avoid reshaping raw traces if reshaping itself is a concern.

o I can understand that the authors refuse to use DL compared to tree models, and I think the “no free lunch theorem” sufficiently justifies this strategy. But in my opinion, the specific arguments cited above are misleading and should be removed from the final version.

At the reviewer’s request, we have now removed the paragraph with our arguments against deep learning methods in our setting from the discussion.

- Authors corrected the name of the python package they used from “shapely” to “Shapely” version 1.7.1. I would like to make sure this is not a mistake: Figure 3 highly resembles plots obtained with the “SHAP” package based on “ShapLEy” (not “ShapELy”) values. Furthermore, SHAP is also cited multiple times by the author. ShapELy v 1.7.1 seems to be a Python package related to geometric analysis rather than feature importance. Can the authors confirm there is no typo here?

We thank the reviewer for correction. Indeed, we are using the SHAP (SHapley Additive exPLanations) python library package to perform feature importance analysis and not the Shapely 1.7.1 library. This typo is now corrected in the manuscript. The following incorrect text "Feature importance was based on the Shapely values from the Shapely 1.7.1 library" has been replaced by "Feature importance was based on the Shapley values from the python package shap version 0.37.0”. 

- Knowing that the authors, even unknowingly, complied with recommendations such as STARD is extremely positive. I think the readership’s trust would highly benefit from citing this in the Methods section, though this is not mandatory.

We thank the reviewer for the suggestion. A citation has now been added to the Materials and Methods section that states that our study aligns with the STARD recommendations for reporting diagnostic studies.

- The table cited in the “Response to reviewers” file, depicting the performance of models according to the M-spike concentration (<1g/L, 1-5g/L, >5g/L) is really interesting, and in my opinion its addition in the manuscript or supplementary material, or at least a sentence in the Results would be of interest for the readers of this work.

We have added a sentence in the Results section as requested.

Reviewer #3: This is an interesting study that demonstrates the analysis of a large set of electrophoresis data using machine learning for the identification of M-proteins in serum. Two referees have thoroughly assessed the paper and provided useful feedback which has been carefully addressed by the authors. While I agree that the ML algorithms used are not overly sophisticated, their applicability to the problem makes the procedure easy to implement. It would be good if this could be tested on 'new' electrophoresis data from various sources to test the transferability of the method, but I am aware this is beyond the scope of the current study. The dataset is extensive and relevant and I think the paper will make a good addition to the literature.

We thank the reviewer for carefully and thoroughly assessing our manuscript and our changes and responses in the first round of revision. We are very grateful that Reviewer #3 took on the task as a new reviewer and appreciate the work the reviewer put in as well as the comments. We hope to access more electrophoresis data in the future from other sources to further test our method as suggested, but as the reviewer already pointed out, this is beyond the scope of the current study.

---

## [Decision Letter · Decision Letter 2]

14 Feb 2024

Machine learning evaluation for identification of M-proteins in human serum

PONE-D-23-17568R2

Dear Dr. Rotter Sopasakis,

We’re pleased to inform you that your manuscript has been judged scientifically suitable for publication and will be formally accepted for publication once it meets all outstanding technical requirements.

Kind regards,

John Adeoye

Academic Editor

PLOS ONE

Additional Editor Comments (optional):

Reviewers' comments:

Reviewer's Responses to Questions

**Comments to the Author**

1. If the authors have adequately addressed your comments raised in a previous round of review and you feel that this manuscript is now acceptable for publication, you may indicate that here to bypass the “Comments to the Author” section, enter your conflict of interest statement in the “Confidential to Editor” section, and submit your "Accept" recommendation.

Reviewer #1: All comments have been addressed

2. Is the manuscript technically sound, and do the data support the conclusions?

Reviewer #1: Yes

3. Has the statistical analysis been performed appropriately and rigorously? 

Reviewer #1: Yes

4. Have the authors made all data underlying the findings in their manuscript fully available?

Reviewer #1: Yes

5. Is the manuscript presented in an intelligible fashion and written in standard English?

Reviewer #1: Yes

6. Review Comments to the Author

Reviewer #1: The authors have carefully updated their manuscript and addressed all interrogations I had in the previous versions, and in my opinion is suitable for publication in its current form.

I would like to thank the authors for their time and for carefully rewriting some parts of their manuscript that may have lacked clarity in the past.

7. PLOS authors have the option to publish the peer review history of their article (what does this mean?). If published, this will include your full peer review and any attached files.

Reviewer #1: **Yes: **Floris Chabrun

---

## [Editor Report · Acceptance letter]

25 Mar 2024

PONE-D-23-17568R2 

PLOS ONE

Dear Dr. Rotter Sopasakis, 

I'm pleased to inform you that your manuscript has been deemed suitable for publication in PLOS ONE. Congratulations! Your manuscript is now being handed over to our production team.

Kind regards, 

on behalf of

Dr. John Adeoye 

Academic Editor

PLOS ONE